# A Levelized Cost Analysis for Solar-Energy-Powered Sea Water Desalination in The Emirate of Abu Dhabi

**Abdullah Kaya [1,*], M. Evren Tok [2] and Muammer Koc [1]**

1   Engineering Department, Hamad Bin Khalifa University, Doha, Qatar; mkoc@hbku.edu.qa
2   College of Islamic Studies, Hamad Bin Khalifa University, Doha, Qatar; etok@hbku.edu.qa
*   Correspondence: akaya@hbku.edu.qa

**Abstract:** The Emirate of Abu Dhabi heavily relies on seawater desalination for its freshwater needs due to limited available resources. This trend is expected to increase further because of the growing population and economic activity, the rapid decline in limited freshwater reserves, and the aggravating effects of climate change. Seawater desalination in Abu Dhabi is currently done through thermal desalination technologies, such as multi-stage flash (MSF) and multi-effect distillation (MED), coupled with thermal power plants, which is known as co-generation. These thermal desalination methods are together responsible for more than 90% of the desalination capacity in the Emirate. Our analysis indicates that these thermal desalination methods are inefficient regarding energy consumption and harmful to the environment due to $CO_2$ emissions and other dangerous byproducts. The rapid decline in the cost of solar Photovoltaic (PV) systems for energy production and reverse osmosis (RO) technology for desalination makes a combination of these two an ideal option for a sustainable desalination future in the Emirate of Abu Dhabi. A levelized cost of water (LCW) study of a solar PV + RO system indicates that Abu Dhabi is well-positioned to utilize this technological combination for cheap and clean desalination in the coming years. Countries in the Sunbelt region with a limited freshwater capacity similar to Abu Dhabi may also consider the proposed system in this study for sustainable desalination.

**Keywords:** desalination; solar power; reverse osmosis; Arabian Gulf; the levelized cost of water

## 1. Introduction

The Emirate of Abu Dhabi is among the seven Emirates which formed The United Arab Emirates (UAE) located in the south-eastern part of The Arabian Peninsula. Abu Dhabi is known to hold a significant amount of oil and gas reserves [1]. Discovery of the oil and gas reserves in the 1960s has propelled The Emirate into a rapid development path which resulted in an exponential increase in population and economic activity [1]. The freshwater need of Abu Dhabi has been increased along this rapid development path which brought the need for seawater desalination. Figure 1 indicates how Abu Dhabi has limited resources when it comes to fresh water, as more than 75% of these freshwater resources have already been depleted [2]. It is clear that Abu Dhabi is no different than the rest of the Arabian Peninsula, which is painted in red color due to excessive depletion of freshwater resources.

## Areas of physical and economic water scarcity

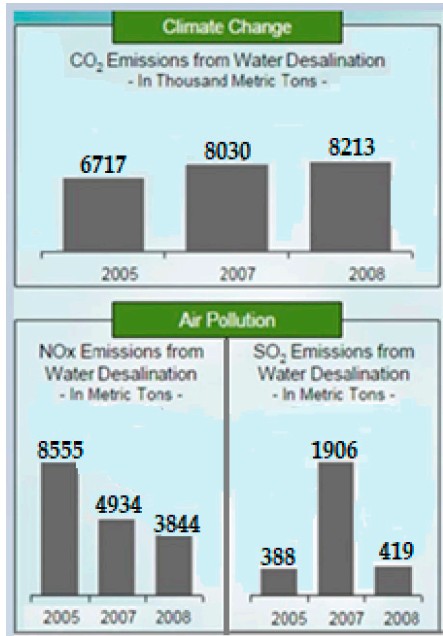

**Figure 1.** The World's Fresh Water Availability by Region. Reproduced from [3].

Most of the desalination in Abu Dhabi is done by using thermal desalination technologies, such as multi-stage flash (MSF) and multiple-effect distillation (MED). More than 80% of the desalination capacity in Abu Dhabi was built using the MSF technology, which was followed by MED as recently as 2008 [4].

These thermal desalination technologies are generally coupled with thermal power plants that use either natural gas or oil for energy production. Due to $CO_2$ emissions and other harmful gases from those thermal power plants, most of the current desalination systems in Abu Dhabi are harming the environment and its citizens [4].

It is clear from Figure 2 that Abu Dhabi's current practice of seawater desalination is not very sustainable considering the fact of these harmful emissions. It is expected that the desalination needs of Abu Dhabi will increase consistently into the 2030s [5].

**Figure 2.** Environmentally Harmful Emissions in Abu Dhabi due to desalination. Modified from [4].

Figure 3 depicts that desalination will play a growing role in the future for Abu Dhabi to meet its water demand. However, the current practice of using MSF and MED technologies for desalination is not a good option regarding the sustainability criteria from environment and health viewpoints. The issue of water scarcity and the need for desalination has actually become a worldwide phenomenon as the demand for cheap and sustainable freshwater escalates rapidly [6]. The adverse effects of climate change in the form of deteriorating air quality, severe droughts, and floods require rapid decarbonization of the energy system, which must include the energy for desalination as well [7]. Therefore, meeting the increasing demand for freshwater through seawater desalination in a clean, cheap, and sustainable way has become a must for such cities Abu Dhabi [8–10].

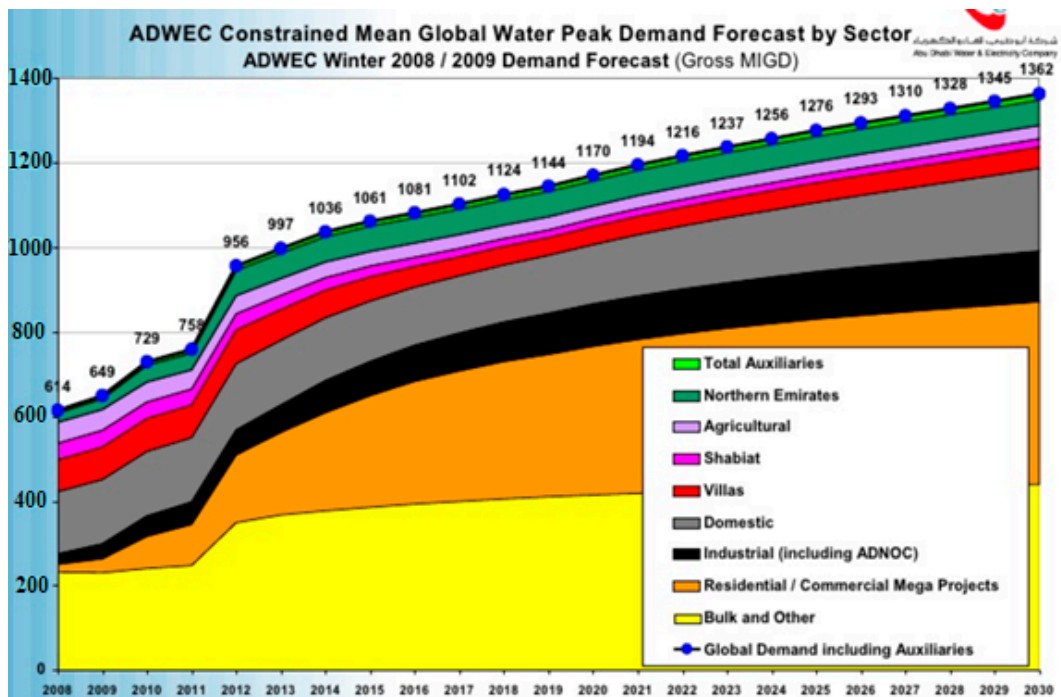

**Figure 3.** Abu Dhabi's Water Demand Projections (in Million Imperial Gallons per Day (MIGD)). Reproduced from [5].

This paper seeks to understand how Abu Dhabi can implement a sustainable desalination scheme by looking at the recent developments in both the desalination and energy sectors. MSF and MED technologies have been in use for more than 50 years, and any further efficiency gains are limited [11]. Both of these technologies require thermal heat, which can be either acquired from the burning of fossil fuels or through concentrated solar or geothermal energy as more sustainable options. The third technology for desalination is called reverse osmosis (RO), which has seen an exponential increase in use recently due to rapid developments in its decreasing cost and increasing effectiveness [12,13]. In most parts of the world, RO has become the main choice for desalination technology as a result [14]. RO systems require direct electricity as the energy form in order to operate during the desalination process as opposed to MSF and MED. Surprisingly, the cost reduction in direct-electricity-producing renewable energy technologies (wind and solar photovoltaic (PV)) has been much sharper compared to the thermal-heat-producing renewables (concentrated solar and geothermal) [13,15]. Considering the convergence of rapid technological developments in both RO for desalination and solar PV, a combination of these two technologies may result in the cheapest and cleanest desalination of the seawater, especially here in the Arabian Gulf region [13].

The second part of the paper explains comparison findings from the literature and the calculations for the specific case of Abu Dhabi. A brief literature survey shows that utilization of RO desalination instead of the current MSF and MED technologies will save Abu Dhabi 10% from fuel consumption.

Implementation of RO instead of the currently established MSF and MED plants would also make a capital investment saving of around 1 billion U.S. Dollars. In the third section, a Levelized Cost of Water (LCW) analysis is conducted for a proposed 90,000 m$^3$/day capacity RO desalination plant run with the electricity from a solar PV system. The findings of this study indicate that an RO system with solar PV as an energy provider could bring a very cheap and clean desalination option to Abu Dhabi. In the last section, results from the analysis and associated limitations are discussed. It is concluded that RO coupled with solar PV seems a very viable solution to Abu Dhabi's sustainable desalination needs.

*Abu Dhabi's Energy and Investment Loss by Due to Current Technological Preferences*

Abu Dhabi's current desalination process is done through co-generation, in which a thermal power plant produces heated steam for electricity and thermal desalination either by MSF or MED [4]. Rapid advancement in energy consumption of RO made the thermal desalination of MSF and MED a costly option. The recent studies indicate that current co-generation systems of electricity production and desalination by MED consume more energy than co-generation of electricity and desalination of water through RO. Precisely, co-generation by MED results in 10% more consumption of oil or gas compared to the separate electricity production and desalination by RO [11].

The findings in Table 1 indicate that co-generation with RO costs 10% less fuel consumption than co-generation with RO for the same amount of water to be desalinated. For 1.79 million m$^3$ of desalinated water, this translates into 536,360 USD or 1,135,790 USD of less cost for the RO option if natural gas (NG) or Fuel Oil is used, respectively. It can be concluded that RO is the preferred option for less energy consumption instead of MED or MSF, which have higher energy consumption than RO [11].

**Table 1.** Comparison of Energy Consumption in Co-Generation Options with multi-effect distillation (MED) or reverse osmosis (RO).

| Co-Generation Options | | | | | | |
|---|---|---|---|---|---|---|
| Scenarios | Cooling Energy Consumption | Desalination | Electricity Production | Fuel Energy Consumption | If NG Is Used | If Fuel Oil Used |
| No Desalination | 302 GWh | 0.00 Mm$^3$ | 63 GWh | 441 GWh | $4,462,920.00 | $9,450,630.0 |
| Co-Generation with MED | 302 GWh | 1.79 Mm$^3$ | 63 GWh | 527 GWh | $5,333,240.00 | $11,293,610.0 |
| Co-Generation with RO | 302 GWh | 1.79 Mm$^3$ | 63 GWh | 474 GWh | $4,796,880.00 | $10,157,820.00 |
| Extra Fuel Cost due to the use of MED instead of RO | | | | 53 GWh | $536,360.00 | $1,135,790.0 |
| Extra Fuel Cost due to the use of MED instead of RO % | | | | | 10.06% | |

NG, natural gas.

Capital expenditures (cap-ex) are a significant cost driver in desalination power plants. The difference in cap-ex for RO, MSF, and MED for the same sized desalination plants are shown in Table 2 [16].

**Table 2.** Capital expenditures (cap-ex) of selected Desalination Technologies.

| Technology | Unit Cost |
|---|---|
| Average MSF Investment Cost | 1750 $/m$^3$-d |
| Average MED Investment Cost | 950.00 $/m$^3$-d |
| Average RO Investment Cost | 800.00 $/m$^3$-d |
| Investment Loss (Conservative Estimate) | 150.00 $/m$^3$-d |

MSF, multi-stage flash.

A calculation has been made for the case of Abu Dhabi's energy overconsumption and investment losses due to the use of MED or MSF instead of RO from a very conservative point of view for the selected years shown in Table 3 (We use MED cap-ex and energy consumption figures to reach

a conclusion for the losses of Abu Dhabi in order to give a conservative estimate. Our findings can be interpreted as minimum losses due to the use of MED or MSF instead of RO).

**Table 3.** Abu Dhabi's Current Energy and Investment Losses due to the use of MED or MSF instead of RO.

| Year/Energy Consumption | | Loss | Water Capacity-1 | Water Capacity-2 | Investment Loss |
|---|---|---|---|---|---|
| 2008 | 77.90 TWh | $73,733,724.71 | 1167.00 MG/d | 4,417,575.55 m$^3$/d | |
| 2009 | 84.40 TWh | $79,886,089.41 | 1243.00 MG/d | 4,705,266.84 m$^3$/d | |
| 2010 | 89.60 TWh | $84,807,981.18 | 1361.00 MG/d | 5,151,945.43 m$^3$/d | |
| 2011 | 95.50 TWh | $90,392,435.29 | 1547.00 MG/d | 5,856,032.02 m$^3$/d | |
| 2012 | 101.50 TWh | $96,071,541.18 | 1585.00 MG/d | 5,999,877.67 m$^3$/d | $854,982,568.16 |

Abu Dhabi decided to use the MED and MSF technologies a long time ago, and these technologies now fall behind RO regarding both energy consumption and investment (cap-ex). While the monetary loss due to energy overconsumption has reached almost 100 million USD ($) per annum, the total investment loss is around 850 million USD for the year of 2012. The argument here is not to convince the Abu Dhabi Government to change those thermal desalination plants to RO, since this would be very costly. However, it is very obvious that RO is a superior technology when it comes to energy consumption and initial capital investment even with the very conservative estimates used in Table 3. The details of a basic thermal plant's service data, the current price of Natural Gas (NG) or fuel oil, and energy conversion tables are provided in Appendix A.

In Section 3, an LCW analysis for desalination is going to be presented. The selected technology for desalination is RO while the energy source will be a solar PV system due to its rapidly decreasing cost [17].

## 2. LCW Analysis: Methodology and Materials

Thanks to the rapid decrease in cost and worldwide government support, solar PV has become an effective way of producing electricity in terms of cost and cleanliness from a fringe technological option 20 years ago. Figure 4 shows the exponential rise in global solar PV installations and associated plummeting costs of the technology [18].

Solar PV has been proved to be a reliable, cheap, and effective energy source that directly utilizes the sun for electricity production. Abu Dhabi is in a very good position to benefit from utilizing the Sun's energy through solar PV technology thanks to its all-year-round and very high solar irradiation [7]. Moreover, the Emirate has already developed local expertise through its Masdar Initiative and various world record projects, such as the cheapest-on-earth Noor 1 350 MW solar PV power plant [19]. The price of electricity from the Noor 1 solar PV power plant will be 2.48 U.S. cents/kWh, which is substantially lower than the cost of electricity in many parts of the world.

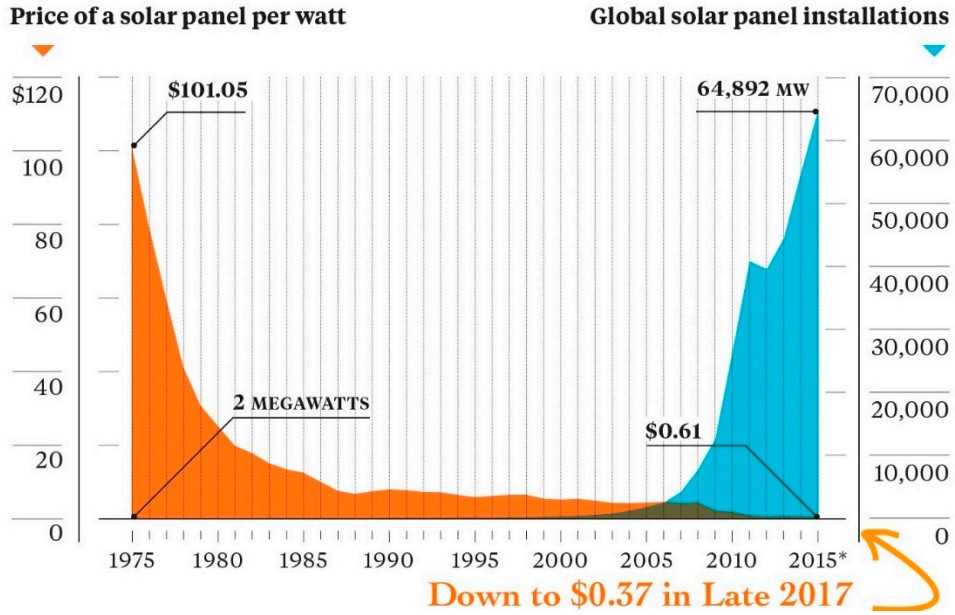

**Figure 4.** Global Solar Photovoltaic (PV) Installation (MW) and the investment cost ($/watt). Reproduced from [18].

Those rapid developments in solar PV technologies and the proven effectiveness of RO for desalination have led to a Solar PV + RO system for sustainable desalination in Abu Dhabi. First, the solar PV electricity output in Abu Dhabi will be calculated based on real data for a given year. This will be followed by an RO system design proposal in order to find the LCW for the Solar PV + RO System in Abu Dhabi.

*2.1. Solar Photovoltatic (PV) Expected Output for a Year in Abu Dhabi*

The output from a solar PV depends on many different factors, such as solar PV cell efficiency, solar radiation (energy) coming from the sun, ambient temperature, and wind speed [20]. The following set of indicators and specific equations for Abu Dhabi were taken from the literature [17]:

- $T_{amb}$: Ambient temperature, $T_{cell}$: Temperature of the PV cell, $T_{stc}$: 25 Celsius
- $I_t$: Solar Radiation on a tilted surface
- IAM: Incident angle modifier
- The ambient wind
- Solar PV cell factor efficiency ($\eta_0$)

We use the following set of equations in order to find the yield from a solar PV system:

1.  Efficiency at standard conditions STC ($T_{stc}$ = 25 °C, Global Horizontal Irradiation (GHI) = 1000 W/m$^2$, Air Mass = 1.5) $\eta_0$: 20%
2.  Temperature Coefficient ($\beta$): $-0.47\%$
3.  Transparency-absorption product ($\tau\alpha_0$): 0.9
4.  Wind Convection factor: $h_0 = 2.8 *$ wind speed $+ 3.0$

In order to find the yield from a solar PV module based on the parameters above, the following set of equations were calculated [17]:

$$\eta = \eta_0(1 - \beta \, T_{cell} - T_{stc})) \tag{1}$$

$$T_{cell} = Tamb + (\tau\alpha_0 * IAM * I_t - I_t * \eta)/(2 * h_0) \tag{2}$$

$$\text{PV yield} = \text{PV efficiency} * \text{It.} \tag{3}$$

A full year of the dataset has been provided by the Mechanical Engineering Department of Masdar Institute for the values in the equations above, which were obtained from a location in Abu Dhabi. Figure 5 and Table 4 highlight expected output (Wh/m$^2$ and kWh/m$^2$) from a 20% efficient ($\eta_0$) solar cell installed in Abu Dhabi with a 10° tilt in Abu Dhabi city.

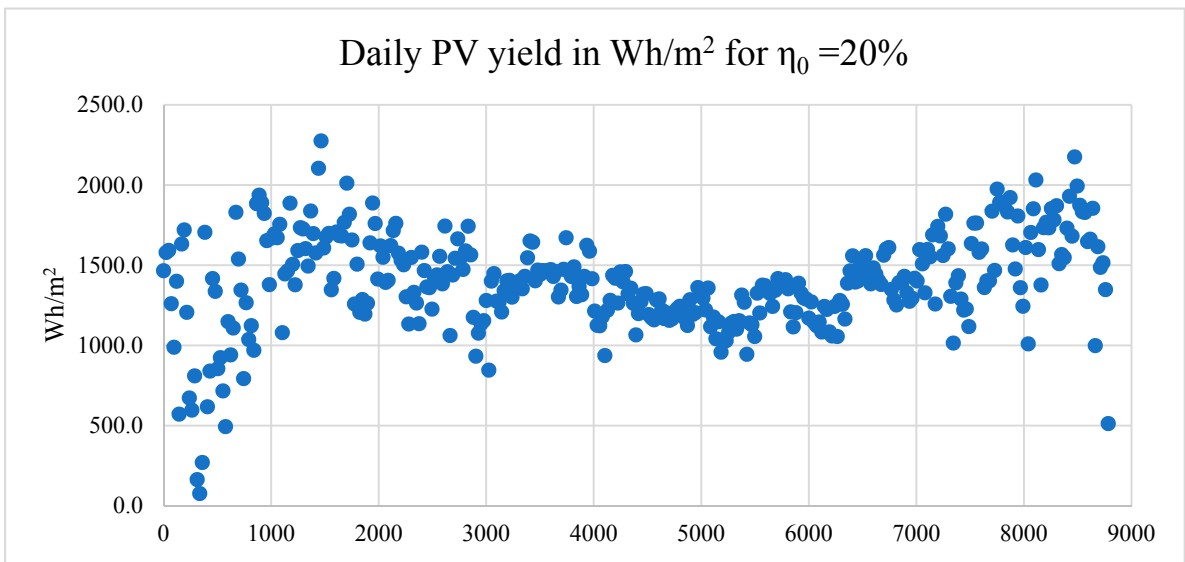

**Figure 5.** Daily Energy Yield from a PV Solar System in Abu Dhabi.

**Table 4.** The Daily and Yearly Energy (electricity) yield from a solar PV system in Abu Dhabi.

| Yield Period | Amount (kWh/m$^2$) |
| --- | --- |
| Yearly yield | 511.08 |
| Average daily yield | 1.40 |

A 20% factory efficient solar PV system yields a daily average of 1.40 kWh of energy per m$^2$, which translates into 511.08 kWh/m$^2$/year as shown in Table 4.

### 2.2. Features of a 90,000 m$^3$/day Seawater RO Desalination System

The membranes of RO were selected from the FilmTech Company owned by Dow Chemicals (Midland, MI, USA), which provides reliable products to the desalination industry. The Dow Filmtec SW30ULE-440i sweater membrane was chosen for this project with the following features:

- 440 square feet membrane surface area
- Max Pressure 1200 psi
- Test Pressure 800 psi
- Minimum Salt rejection at 700 psi 99.6%
- Stabilized salt rejection at 700 psi 99.7%
- Permeate flow 12,000 gpd at 700 psi w/32,000 ppm NaCl feed

The membrane has a current price of $820 (45 m$^3$/day capacity) with 5 years of expected lifetime [11]. Regarding the RO system, the following design was selected with ultra-filtration (UF) as the pretreatment method as shown in Figure 6 [21].

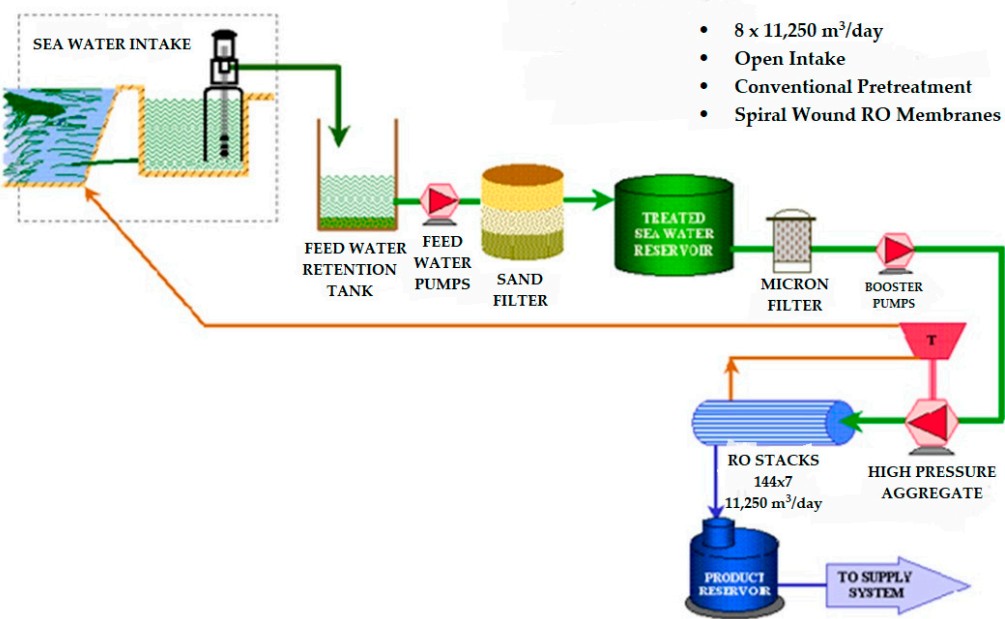

**Figure 6.** The RO Desalination System. Reproduced from [21].

The cost of the RO system, excluding the membrane, such the UF unit, pressure exchanger, and media filtration, is adapted from [11] with the assumption that there was a 1% cost reduction in the system components from 2003 to 2016 following the industry trends [21].

*2.3. Capital Expenditure (Cap-Ex) and Levelized Cost of Water Estimates for the Solar PV + RO System*

We made the following conservative assumptions in order to calculate an upper bound of capital expenditure (cap-ex) and LCW from a solar PV + RO system:

- The land is given free for both the RO and Solar PV facility;
- The permitting costs in Abu Dhabi for this project would be between 5% and 10% of total cap-ex;
- A zero-balance electricity agreement is assumed with the Abu Dhabi Water and Electricity Authority (ADWEA), meaning that excess electricity from the solar PV system will be given to the grid during the day and bought back during the night time;
- The project has a lifetime of 25 years, where the solar PV cells operate with 1% efficiency loss per year. The system-wide losses, such as transmission or inverter losses, are ignored due to a practical purpose;
- The solar PV system is conservatively set to cost 200 $/m$^2$ (1 $/watt multiple of the solar cell price), which is higher than current industry practices (150 $/m$^2$ → 0.75 $/watt). That includes the cost of the inverter and other equipment, such as the steel structure;
- The RO system consumes 3.5 kWh of electricity per m$^3$ of water desalinated;
- The annual cost components are chemicals (1.4 million $/year) and overhead cost (313,000 $/year). These values are obtained from a 2003 study for the same capacity desalination plant [21]; and
- Project financing is assumed to be 5%, which is well above the current interest rates.

The basic capacity, number of membranes, and the required solar PV system are summarized in Table 5.

**Table 5.** The Proposed Solar PV + RO System's Basics.

| System Feature | Amount |
|---|---|
| Plant Capacity (m$^3$/day) | 90,000 |
| Number of RO membranes needed | 2000 |
| Solar PV System Size (MW) | 1124.825 |

All other cost data for the system components (excluding membranes and including the annual chemical cost and overhead cost) were taken from an actual study conducted in 2003 [12]. Therefore, assumptions for the RO system are highly conservative and the actual current cost values could be well below what is assumed in this study. In order to deliver a variety of industry dynamics and to account for rapid development in both Solar PV and RO technologies, a set of different cases is proposed based on expected development in both of these industries. Table 6 summarizes these different cases proposed in the calculation of cap-ex and LCW for the proposed Solar PV + RO system.

**Table 6.** Sensitivity Analysis of the levelized cost of water (LCW) for the PV + RO System.

| | RO System Cost Decrease %/Year | Solar PV Cost ($/m²) | Permit Cost as % of Total Cap-Ex | RO Energy Consumption kWh/m$^3$ |
|---|---|---|---|---|
| 1. Baseline Case | 1% | 200 | 5% | 3.5 |
| 2. Low-Cost (LC) RO | 2.5% | 200 | 5% | 3.5 |
| 3. LC RO + High Permit Cost (HPC) | 2.5% | 200 | 10% | 3.5 |
| 4. LC RO + HPC + Low-Cost PV | 2.5% | 150 | 10% | 3.5 |
| 5. LC RO + HPC + LC PV + Low-Energy RO | 2.5% | 150 | 10% | 2.5 |

## 3. Results and Discussions

There are in total five different cases to evaluate in order to calculate the cap-ex distribution of the system and LCW. The baseline case assumes a 1% cost decrease for the RO system for the years between 2000 and 2013, whereas the cost of a solar PV system is 200 $/m². The permit cost is assumed to be 5% of total cap-ex, and RO energy consumption is assumed to be 3.5 kWh/m³. The cap-ex value and its distribution among system components for the first case are shown in Figure 7.

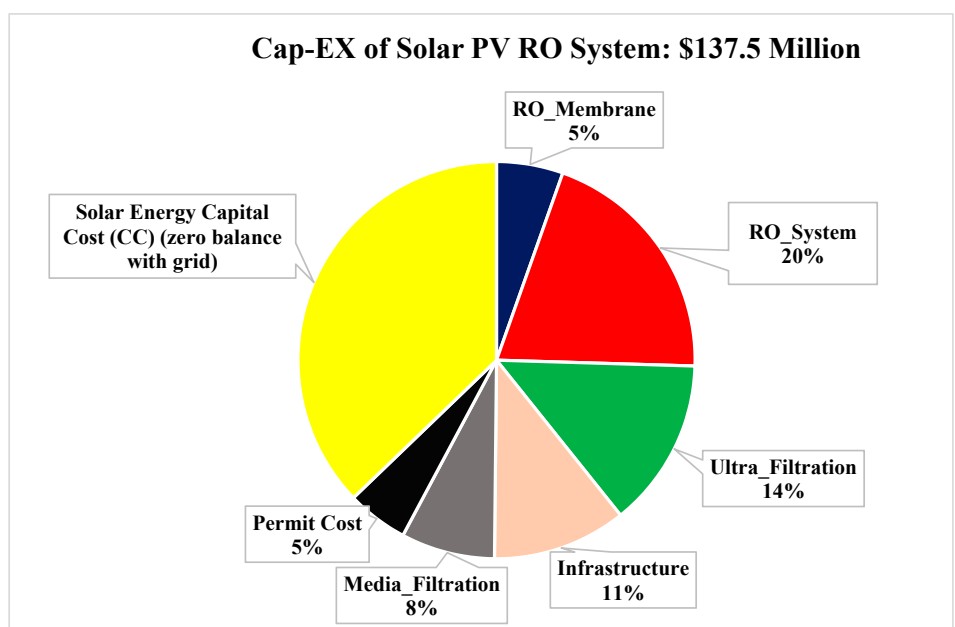

**Figure 7.** Cap-Ex Distribution for the First Case.

It is clear that the total RO system, including membranes, other components, and pretreatment units, comprises the bulk of cap-ex (58%). The suggested solar PV system would require an investment

totaling to 37% of the whole cap-ex. The unit investment cost is 1527 $/m$^3$ in this scenario, which is in an acceptable range when considering the fact that the plant will have no electricity cost later.

In the second case, we assume that the RO system cost may have decreased 2.5% per year instead of the 1% assumed in the baseline case, while Solar PV cost, permit cost, and RO energy consumption are the same as the baseline case. The cap-ex value and its distribution among system components are shown in Figure 8.

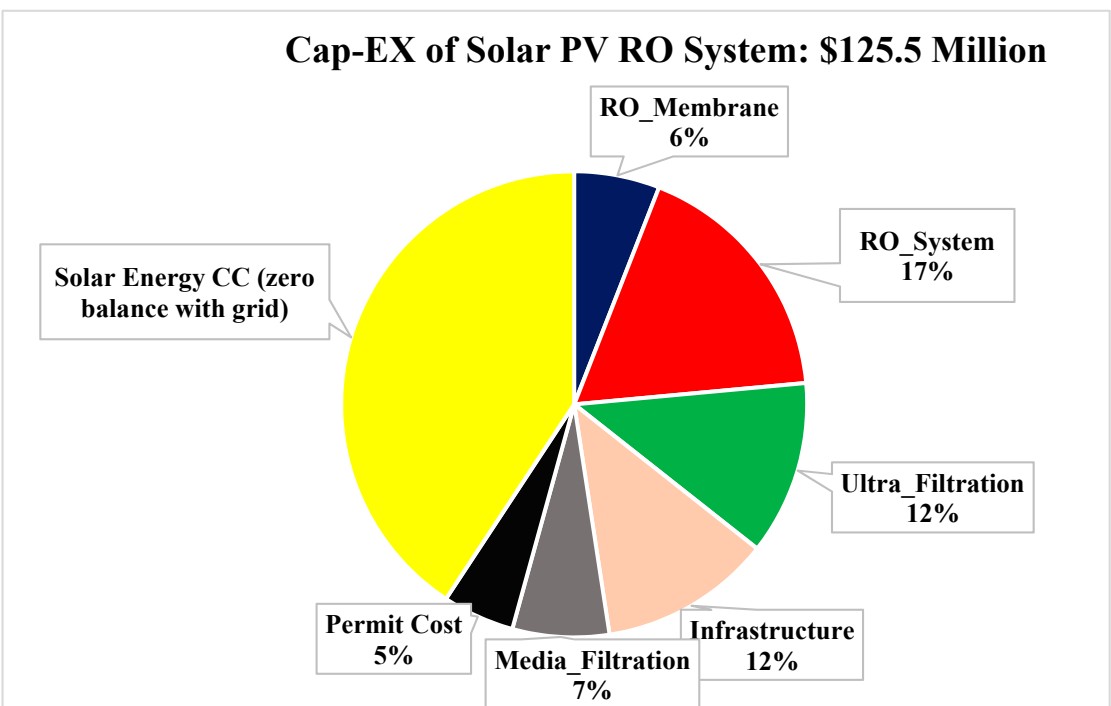

**Figure 8.** Cap-Ex Distribution for the Second Case.

The decreasing cost of RO desalination system components, which comprise pressure pumps, membrane cases, and pretreatment units, resulted in a decreasing share of the RO desalination system in cap-ex (54%). The suggested solar PV system would require an investment totaling to 41% of the whole cap-ex, while its absolute cost remains the same. The unit investment cost is 1395 $/m$^3$ in this scenario, which is lower than the investment cost of the first case.

In the third case, permit costs are assumed to be high (10% of total cap-ex), following the recent trend in the desalination industry. The decrease in RO system cost is assumed to be 2.5% and there is no change in solar PV cost and RO energy consumption as for the second case. Figure 9 highlights the results for the cap-ex of the third case.

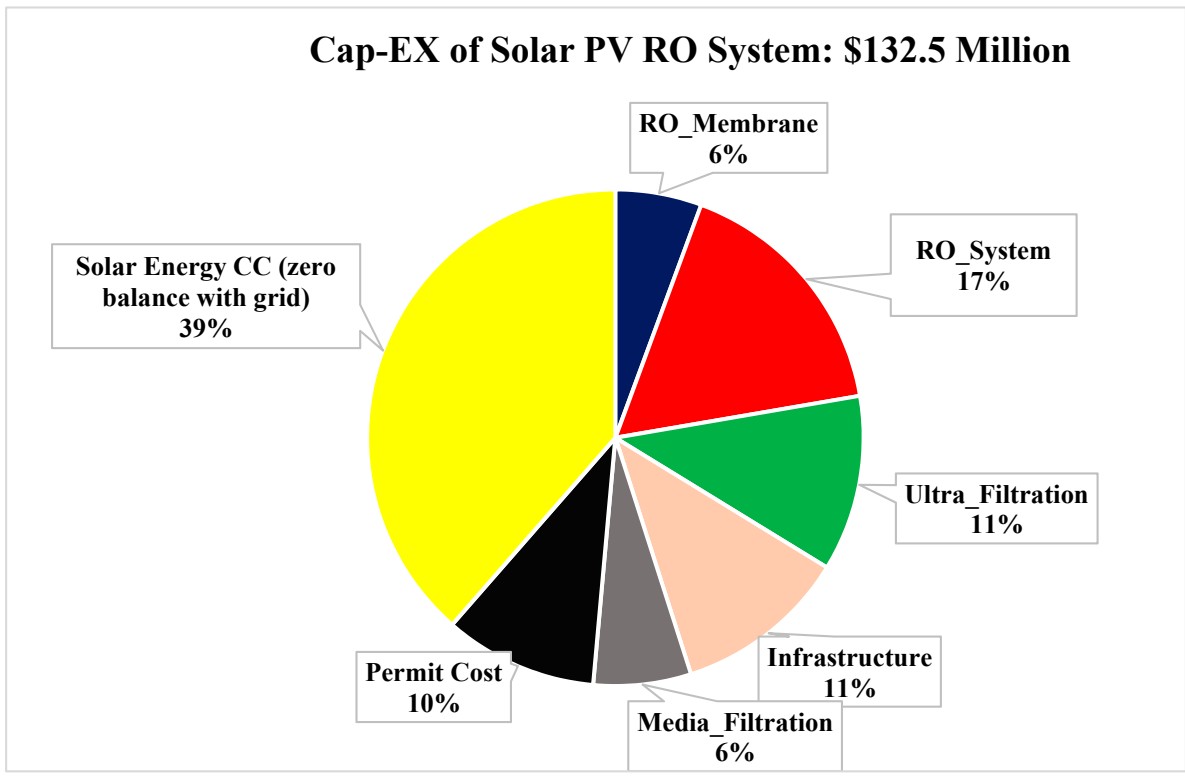

**Figure 9.** Cap-Ex Distribution for the Third Case.

Recent trends indicate that permit costs are likely to increase and take a significant share from the investment budget. If the permit costs reach 10% of the total investment cost, then this will bring an additional $5 million of the cost compared to the second case. The unit investment cost is 1472 $/m$^3$ in this scenario, which is higher than the investment cost of the second case.

In the fourth case, solar PV system installation cost is assumed to be 150 $/m$^2$, following the world record projects of the sector in the region and particularly in the UAE. The decrease in RO system cost with 2.5%, high permit costs (HPC) as 10% of the total project, and RO energy consumption of 3.5 kWh/m$^3$ are the same as for the third case.

Solar PV energy has shown rapid technological improvement and a large decrease in costs for modules and other system components as discussed above. A 25% cost decrease compared to the baseline case is assumed here. This assumption may have actually been achieved in current solar PV projects. The results in Figure 10 indicate that decreasing solar PV costs will greatly contribute to lower cap-ex values for solar-PV-powered RO desalination projects. The reason is clear, as solar PV was comprising the largest share in cap-ex for the previous cases.

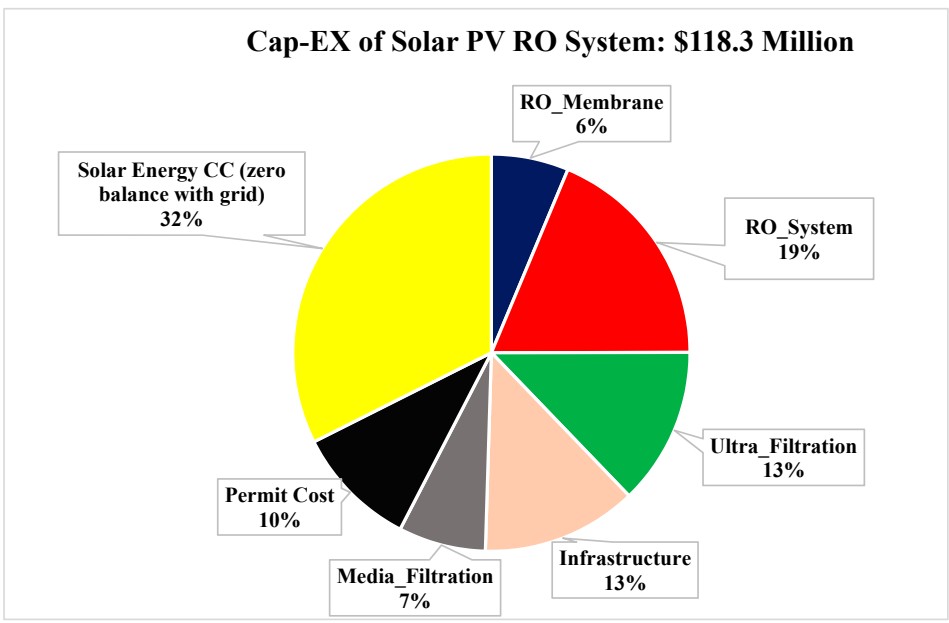

**Figure 10.** Cap-Ex Distribution for the Fourth Case.

In the fifth and final case, energy consumption of RO is assumed to be 2.5 kWh/m$^3$, while the RO system cost decrease, the solar PV cost, and the permit cost are assumed to be same as in the fourth case. The research and development have intensified in the RO industry due to the competitive pressure from the market. As a result, the energy consumption of RO for seawater desalination is expected to decrease further and may even go below 2.5 kWh/m$^3$. It is indeed reported that 'other new energy recovery systems are under development with the objective of reducing energy consumption for seawater RO (SWRO) below 2 kWh/m$^3$ [13]. The results below indicate that the cap-ex value of solar-PV-powered RO will decline even further with an efficiency increase in energy consumption as shown in Figure 11.

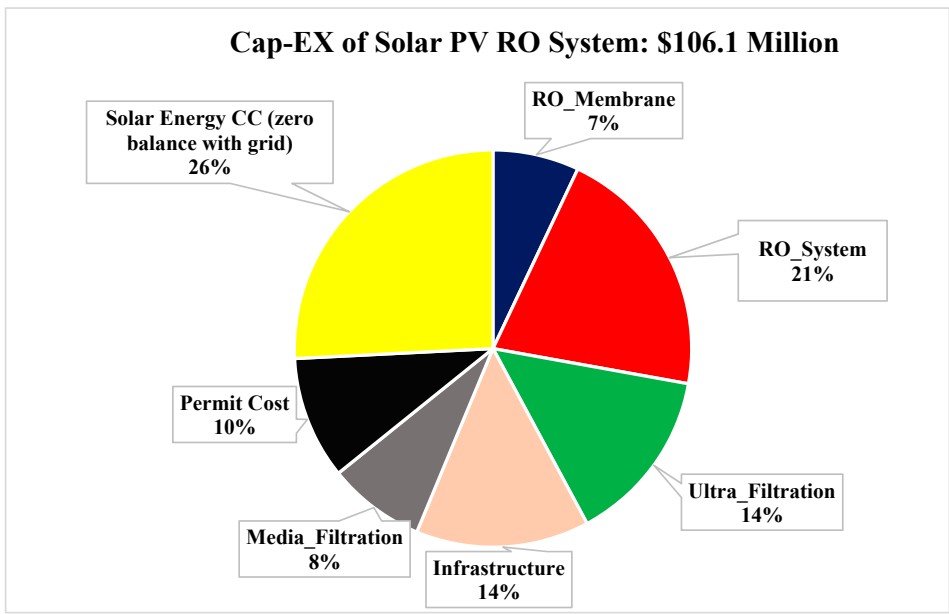

**Figure 11.** Cap-Ex Distribution for the Fifth Case.

In order to calculate the LCW for each case, the following assumptions are made:

- The project financing interest rate ($i_t$) is 5%, which is conservative considering the prevailing interest rates (2~4%);
- The annual chemical cost is $1.4 million following [22];
- The annual overhead cost is $330,000 following [22];
- The decommissioning cost of the project ($F_T$) is assumed to be 0;
- The project lifetime ($T$) is 25 years; and
- The solar PV system's energy output is assumed to remain constant.

The following general formulas are applied to calculate the LCW of the proposed Solar PV + RO desalination system [23]:

$$\text{LCW } (\$/\text{m}^3) = \frac{C_0 + \sum_{t=1}^{25} \frac{OM_t + F_t}{(1+i_t)^t}}{\sum_{t=1}^{25} \frac{W_t}{(1+i_t)^t}}$$

where $C_0$ represents cap-ex, $OM_t$ represents annual costs (chemical and overhead), and $F_t$ represents decommissioning costs. The project lifetime is 25 years and the interest rate of financing ($i_t$) is assumed to be 5%, which is well above the current interest rates in the UAE (3%).

The results of the LCW for each case are as follows:

Appendix B provides the baseline case Excel table to calculate the cap-ex and LCW for the proposed solar-PV-powered RO desalination system. It is clear from Figure 12 that the LCW of the proposed solar PV + RO desalination system is well below the recent estimates of other desalination technologies or RO without a solar PV system. Table 7 shows a recent LCW estimate (under the total cost in the year 2013) of desalination technologies that run without a solar PV system [24]:

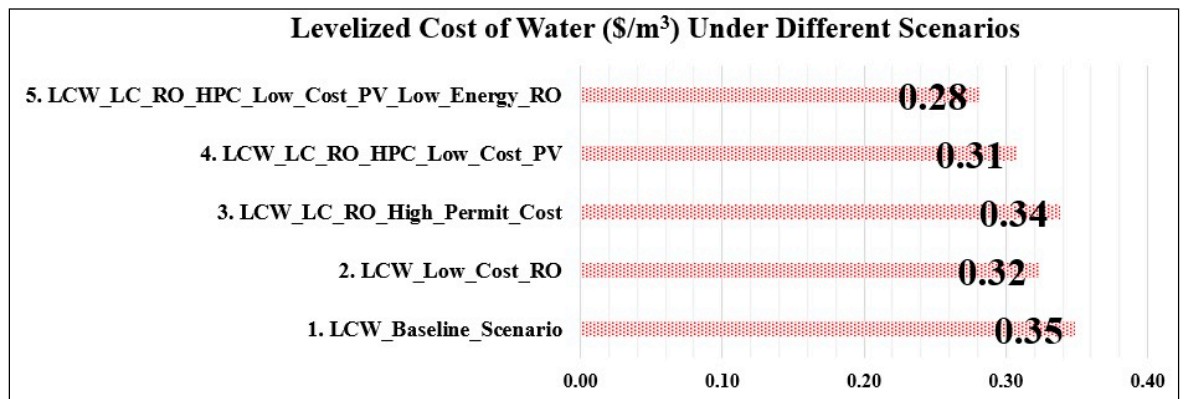

**Figure 12.** The LCW Estimates Under Each Scenario.

**Table 7.** The Energy Consumption and LCW Estimates of Desalination Technologies. SWRO, seawater RO; BWRO, Brackish water RO.

| Process | Thermal energy kWh/m$^3$ | Electric energy kWh/m$^3$ | Total energy kWh/m$^3$ | Investment cost $/m$^3$/d | Total water cost US$/m$^3$ |
|---|---|---|---|---|---|
| MSF | 7.5–12 | 2.5–4 | 10–16 | 1200–2500 | (0.8–1.5)[a] |
| MED | 4–7 | 1.5–2 | 5.5–9 | 900–2000 | 0.7–1.2 |
| SWRO | - | (3–4)[b] | 3–4 | 900–2500 | 0.5–1.2 |
| BWRO | - | 0.5–2.5 | 0.5–2.5 | 300–1200 | 0.2–0.4 |

[a] Including subsidies (price of fuel); [b] Including energy recovery system.

It is clear that the proposed solution in this paper has lower LCW estimates from the seawater RO (SWRO) LCW estimates from the selected paper. This difference in LCW estimates of our own and the selected study may be surprising at first glance. Nonetheless, the key reason for this difference is the rapid decrease in the cost of solar PV in which the Emirate of Abu Dhabi has abundant solar energy radiation. Considering the fact that Abu Dhabi will pay only 2.48 U.S. cents/kWh to electricity from a solar PV plant to be built soon, our results can make more sense.

## 4. Conclusions

Seawater desalination will play a significant role in the Emirate of Abu Dhabi in the coming years due to the growing population and economic activity in the face of declining freshwater resources. Sustainable desalination of seawater regarding a clean energy resource and economical technology option is a must for Abu Dhabi to meet its vision 2030 targets. The currently prevalent desalination technologies MSF and MED are causing the emission of harmful gasses, such as $CO_2$ and $NO_x$ derivatives, to the atmosphere. It has also been shown that these thermal desalination technologies consume at least 10% more fuel than RO-based desalination technologies. As a result, Abu Dhabi has paid in excess of 100 million U.S. $ in 2012 due to its choice of MSF and MED over RO to fuel expenses for seawater desalination. RO technology has another advantage over its thermal competitors regarding the cap-ex costs due to the rapid technological advancement of the former. Based on the 2012 installed desalination capacity in the Emirate, a minimum loss of 800 million U.S. $ occurred due to the preference of MSF and MED over RO for seawater desalination.

An alternative sustainable desalination method for Abu Dhabi has been analyzed by considering solar-PV-powered RO seawater desalination. The rapid cost decrease in PV technology and Abu Dhabi's all-year-round high solar radiation level make solar PV an ideal candidate for a clean and cheap form of energy production. A one-year study indicates that a solar PV system with 20% efficiency will yield 511 $kWh/m^2/year$ in Abu Dhabi at a system of cost of 200$/$m^2$. Land for this system installation is assumed to be given free. Zero-Balance access to the grid is assumed, which means that the solar PV system will give its extra energy to the grid during the day and take the same amount of energy from the grid at night with no monetary transaction.

A 90,000 $m^3$/day seawater RO desalination system has been devised following a similar study conducted in 2003 [12]. RO membranes were selected from FilmTech Company with features best suited to the conditions of seawater in Abu Dhabi, such as strong flux under high salinity conditions. Other system components and their average prices, including operation and maintenance costs, were obtained from the aforementioned reference study [21].

Our analysis indicates that cap-ex costs will be around 137.5 million U.S. $ (1527 $/$m^3$) under the most conservative baseline case with an LCW of 0.35 $/$m^3$. Another four cases were devised in reference to the baseline case considering each one of the following developments:

- a higher decrease in RO system component costs with respect to the reference study in line with the developments in the technology since 2000;
- expected high permit costs;
- falling solar PV costs; and
- lower energy consumption of the RO system.

It has been shown that the cap-ex cost of the solar-PV-powered RO system has the potential to go as low as 106 million U.S. $ (1179 $/$m^3$) when all the potential developments listed above are taken into account at the last (fifth) case. The LCW values fluctuate between 0.35 $/$m^3$ for the baseline case and 0.28 $/$m^3$ for the best (fifth) case. These findings are then compared to the recent reports from the literature of existing desalination systems [13]. Even the baseline case with highly conservative assumptions is cheaper than 100% from all thermal desalination methods regarding the cap-ex per capacity and LCW. The exponential cost decreases in both the solar PV and RO sectors along with the increasing economic scale globally signal the fact that cheap and clean desalination can be achieved by the combination of these technologies. Furthermore, these findings do not just point to a solar-powered RO desalination future for Abu Dhabi only but also for the Gulf and greater Middle East in general, which have plentiful sun energy throughout a year.

The substantially lower LCW values have become possible thanks to the rapid decrease in costs of both RO and solar PV technologies [25,26]. A study has found that the average LCW of a solar-PV-powered RO desalination system to be between 2 and 3 US $ per $m^3$ in 2009, which is, at minimum, 6 times the values we obtained in this study [27]. Particularly, the solar PV system cost

has exponentially dropped in the last two decades and it continues to drop further [28]. The cost of producing electricity from a solar PV system has already become the cheapest among all energy sources in many parts of the world [29]. These rapid cost reductions in both solar PV and RO technologies are expected to accelerate thanks to economies of scale and continuous innovation, which will strengthen the argument for their use for sustainable desalination [30].

This study has made some strong assumptions that may slow the envisaged potential of solar-PV-powered RO desalination in the Emirate of Abu Dhabi. First of all, the proposed solar PV system with a 1.1 GW capacity would need land a minimum of 5 km$^2$ in size. This was assumed to be given free, which would be difficult due to the growing number of solar PV projects in Abu Dhabi. Another concern for the solar PV energy is its intermittent nature, which causes difficulties in the management of the whole electricity grid. Therefore, the increasing utilization of solar PV for both energy production and desalination will put significant pressure on the Emirate concerning grid management. Regarding RO desalination, the ultra-filtration (UF) method is assumed to be working well for the pretreatment in this study, which may not always be the case [14]. The highly saline water of the Arabian Gulf and the region's other idiosyncratic sea features may cause some serious problems during operation, such as fouling and scaling [31]. Apart from these considerations, there are other forms of sustainable desalination methods to be taken into account for a better comparison and greater understanding. MED is a cheap and effective form of thermal desalination with a significant presence in Abu Dhabi. Concentrated solar power (CSP) produces thermal energy by utilizing radiation from the Sun. Abu Dhabi is suitable for CSP technology to produce energy as there is already a 100-MW capacity CSP plant operating in the Emirate. A combination of CSP with MED for sustainable desalination can be conducted in the future to compare the solar PV with RO system proposed in this study.

**Author Contributions:** Methodology, analysis, writing—original draft preparation, A.K.; writing—review and editing, M.E.T; writing—review and editing, M.K.

**Funding:** The authors declare that no external funding has been taken during the publication of the article.

**Acknowledgments:** We acknowledge the financial support provided by both the Masdar Institute of Science and Technology (part of Khalifa University) and the Hamad Bin Khalifa University (HBKU). The publication of this article was funded by the Qatar National Library.

**Conflicts of Interest:** The authors declare no conflict of interest.

## Appendix A

Thermic Plant Service Data and Energy Conversion Tables [11].

| Thermic Power Plant Service Data | Value | Cost |
|---|---|---|
| Steam Inlet Temperature | 540 °C | - |
| Steam Inlet Pressure | 160 Bar | - |
| Steam Outlet Temperature | 320 °C | - |
| Steam Outlet Pressure | 40 Bar | - |
| Turbine Speed | 12,000 r/min | - |
| Steam Production | 15 kg Steam/kWh | - |
| Energy Consumption | 3600 kJ/kWh | - |
| If NG is used | 0.57 m$^3$NG/kWh | 10.12 $/MWh |
| If Fuel Oil is used | 1 kg Fuel Oil/kWh | 21.43 $/MWh |

| Fuel | Specific Energy (Wh/kg) | Energy Density (Wh/L) | Price ($/MT) | Cost ($/MWh) |
|---|---|---|---|---|
| NG | 15,416 | 10 | 156 | 10.12 |
| Diesel | 13,333 | 9944 | 550 | 41.25 |
| Fuel Oil (IFO 380) | 13,066 | 10,143 | 280 | 21.43 |
| Coal (bituminous) | 6667 | 7222 | 93 | 13.95 |

## Appendix B

Capital expenditures and Levelized Cost of Water for the Baseline Scenario.

| | |
|---|---|
| Project Life Time (years) | 25.00 |
| The capacity of Membrane System (m$^3$/d) | 90,000.00 |
| Membrane Life Time (years) | 5.00 |
| Project Financing Cost (%) | 5.00 |
| Dow Filmtec SW30ULE-440i Seawater Membrane Flow (m$^3$/day) | 45.00 |
| Unit Price of Membrane ($) | 820.00 |
| Number of RO membranes needed | 2000.00 |
| RO_Membrane | 7,455,358.83 |
| Ro_System | 27,563,183.19 |
| Ultra_Filtration | 18,949,688.45 |
| Infrastructure | 15,000,000.00 |
| Media_Filtration | 10,508,463.59 |
| Permit Cost | 6,551,036.69 |
| Solar Energy CC (zero balance with grid) | 44,993,003.05 |
| Total Capital Cost ($) | 131,020,733.80 |
| Ultra_Filtration_Chemical_Cost ($/m$^3$) | 0.03 |
| Total Treatment Energy Consumption (kwh/m$^3$) | 3.50 |
| Annual Energy Consumption (kwh/year) | 114,975,000.00 |
| Solar Energy System Required (zero balance with grid) (m$^2$) | 224,965.02 |
| Unit Solar PV system Cost for 20% ($/m$^2$) | 200.00 |
| Chemicals Cost ($/year) | 1,400,000.00 |
| Overhead (S/year) | 313,000.00 |
| LCW ($/m$^3$) | 0.34 |

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
