# Peer review of "A Levelized Cost Analysis for Solar-Energy-Powered Sea Water Desalination in The Emirate of Abu Dhabi"

_sustainability, doi:10.3390/su11061691_

Reviewer 1 Report

The topic is interesting and it is adapt to this journal. There are few comments and/or suggestions to improve the manuscript.

-The following structure would be preferable based on the Sustainability Microsoft Word template file: 1. Introduction (1.1, 1.2, 1.3.), 2. Materials and Methods (2.1, 2.2., 2.3.), 3. Results and Discussion (3.1, 3.2, 3.3), 4. Conclusions. These sections mixed in the text.

https://www.mdpi.com/journal/sustainability/instructions

-p1, 31: ‘reserves in the 1960s has propelled’

--The point is missing at the end of the sentence.

-Figures 1-5, 7.

--Please add references to Figures 1-5, 7. Please improve the image quality of Figures 1,2,5,7. 

--Please change the CO2, m2 and m3 abbreviations to CO2, m2 and m3. Please check the full manuscript. 

--Please optimize Figure 3. The size of the text is too large. See the bottom left of the image. 

--The units in Table 1 were not properly edited, please check.

-PV panel words.

-Are you thinking PV module? The PV panel word is inappropriate. Suitable for example: PV modules, photovoltaic modules, solar PV modules, etc. The ‘solar panel’ means different. Please check the entire text. Easier to understand here:

https://www.mdpi.com/2079-9292/8/2/149

-p7, 156, Tstd

-- Do you mean Tstc? Please check the entire text.

-p7, 171, ‘ 1. Efficiency at standard conditions STC(Tambient = 25°C, GHI = 1000W/m2) η0:20%’

-- I cannot find the STC information about the air mass 1.5 (AM1.5), please expand.

-p7, 176

-- The absorption value is too high. Check the absorption values in this manuscript:

-- Check also the convection coefficient from this manuscript:

https://www.mdpi.com/1996-1073/11/5/1114/htm

--Please search references to the equations. Where did you find the equations? Without quotation it is considered plagiarism.

-Figure 6.

--Please add unit to Figure 6.

-p13, 305

--This relationship is unreadable, please edit again.

-Figure 6, 'Daily energy Yield in wH/m2’

--The correct marking is the following: Wh / m2

-p7,170-172.

--These equations apply to PV cells but you mentioned also PV panel and PV system. These equations are not suitable for calculating the energy production of the PV system. Where can we found for example the PV system losses? This part should be recalculated.

https://photovoltaic-software.com/principle-ressources/how-calculate-solar-energy-power-pv-systems

--It would be better to use the JRC Photovoltaic Geographical Information System (PVGIS) software. It is much easier to determine the value of PV yield under real climatic conditions.

http://re.jrc.ec.europa.eu/pvg_tools/en/tools.html#PVP

http://re.jrc.ec.europa.eu/pvgis/apps4/pvest.php?map=africa&lang=en

Here are some aspects of the use of PV system and PVGIS (p.8)

https://www.mdpi.com/2079-9292/8/2/149

-p9, 214-215: ‘The project has a lifetime of 25 years where the solar PV cells operate without any efficiency loss’

--This is not correct. You need to calculate with PV system losses and PV module degradation. In addition, PV inverter replacement should also be considered every 10-15 years.

https://onlinelibrary.wiley.com/doi/abs/10.1002/pip.1182

-Figure 8

--Please use same colors in the case of Figures 8-12.

Author Response

Response Reviewer #1

We would like to thank the first reviewer for his insightful and constructive comments. We have revised the paper based on the comments and suggestions of the two reviwers extensively. The editions can be seen in yellow highlight. 

1-    The structure of the paper is changed based on the suggestions of the reviewer.

2-    Lines from 30 to 33 are rewritten to fix the English.

3-    References are added to figures 1, 2, 4, and 6. The figure 2 in the previous draft has been removed since there was no version of it with a better resolution. The resolutions of other figures are enhanced accordingly.

4-    All the changes were done to the units such as CO2, m2 and m3 throughout the paper.

5-    Figure 2 (previously Figure 3) has been optimized based on the suggestion.

6-    The units in table 1 are properly edited.

7-      “PV panel” is replaced by “PV module” throughout the entire text

8-    Tstd is replaced with the correct form Tstc

9-    Absorption and convection values are obtained from the study where the data is made available as well (Mokhtar et al. 2010). We agree with the reviewer for the fact that these values may be slightly over but won’t substantially change the results. We prefer to keep the current values. The data we used to calculate the solar PV module output in Abu Dhabi contains the whole year measurement of wind, radiation, temperature. We would like to stick to the current methodology in calculating the solar PV module output.

10- All the equations are given with the reference as suggested by the reviewer

11- Unit is added to Figure 5 (previously Figure 6)

12-  Lines from 301 to 315 are rewritten to clarify the way we calculated LCW for the proposed RO + Solar PV system

13- The reviewer rightfully suggested that we should consider the degradation of the solar PV modules in their assumed 25-year lifetime. We now assume that solar PV modules lose 1% per year from their efficiency level. All the calculations are updated accordingly. The annual cost of the solar PV system as well as total system cost are calculated conservatively which include the inverter cost as well.

14- Figures from 7 to 11 are unified in their colors.

Mokhtar, Marwan et al. 2010. “Systematic Comprehensive Techno-Economic Assessment of Solar Cooling Technologies Using Location-Specific Climate Data.” Applied energy 87(12): 3766–78.

Reviewer 2 Report

This paper emphasizes the superiority of reverse osmosis (RO) to multi-stage flash (MSF) and multi-effect distillation (MED) through conducting a levelized cost calculation. The logical flow of this paper is as follows. The combination of RO with a renewable energy technology, photovoltaic (PV), offers more efficient desalinations because it consumes less energy and emits smaller amounts of CO2 and other harmful substances.

 As far as the manuscript was read, the levelized cost analysis seems to be made based on the data sources which can be mostly reliable. The logical flow drawing the conclusion is very clear. More importantly one of the most important issues is dealt with by this study. This study thus, has a high potential to be accepted. To brush this paper up furthermore, I’d like to recommend the authors to additionally address the following respects.

(1) Some researchers and engineers have pointed out disadvantages (such as total dissolved salinity) specifically lying in RO. In this paper, the disadvantages of MSF and MED only are highlighted. To construct a fair and objective analysis, this paper should take into account the disadvantages of RO as well.

(2) Is the concept of combining the RO with PV proposed by the authors? If not, refer to relevant previous works in Introduction.

(3) Refer to Tables 2 and 3 in the text.

(4) Line 104-105: …that co-generation with RO costs 10% less fuel consumption than co-generation with ?.

(5) Line 135: That chart below (Fig. 5) shows

Author Response

Response Reviewer #2

We would like to thank the second reviewer for his insightful and constructive comments. We have revised the paper based on the comments and suggestions of the two reviewers extensively. The editions can be seen in yellow. 

1-    We agree to the fact that RO has some disadvantages as well such as fouling and scaling. We mention these from lines 395 to 401 and further discuss consideration of MES or MSF as well in the conclusion.

2-    We as authors propose the concept of combining RO with solar PV.

3-    We refer to tables 2 and 3 in the main text as suggested by the reviewer

4-    Lines 104-105 in the previous version has been rewritten for clarification (now lines 106-107).

5-    Lines 137 to 138 are rewritten.

Round  2

Reviewer 1 Report

This revised manuscript has addressed all my previous concerns. Therefore, I recommend that this paper in its current form is acceptable to the journal. Nevertheless, I think that equations 1-3 are not the most accurate. This reduces the scientific accuracy and reliability of the manuscript.

Reviewer 2 Report

No further revisions are required because the present revision well responded to the review comments and is improved enough.